# Graph Convolutional-Based Deep Residual Modeling for Rumor Detection on Social Media

**Na Ye [1], Dingguo Yu [2,3,*], Yijie Zhou [3,4], Ke-ke Shang [5,*]** and **Suiyu Zhang [3]**

[1] School of Journalism and Communication, Communication University of Zhejiang, Hangzhou 310018, China
[2] College of Media Engineering, Communication University of Zhejiang, Hangzhou 310018, China
[3] Key Lab of Film and TV Media Technology of Zhejiang Province, Hangzhou 310018, China
[4] Institute of Intelligent Media Technology, Communication University of Zhejiang, Hangzhou 310018, China
[5] Computational Communication Collaboratory, Nanjing University, Nanjing 210023, China
* Correspondence: yudg@cuz.edu.cn (D.Y.); kekeshang@nju.edu.cn (K.-k.S.)

**Abstract:** The popularity and development of social media have made it more and more convenient to spread rumors, and it has become especially important to detect rumors in massive amounts of information. Most of the traditional rumor detection methods use the rumor content or propagation structure to mine rumor characteristics, ignoring the fusion characteristics of the content and structure and their interaction. Therefore, a novel rumor detection method based on heterogeneous convolutional networks is proposed. First, this paper constructs a heterogeneous map that combines both the rumor content and propagation structure to explore their interaction during rumor propagation and obtain a rumor representation. On this basis, this paper uses a deep residual graph convolutional neural network to construct the content and structure interaction information of the current network propagation model. Finally, this paper uses the Twitter15 and Twitter16 datasets to verify the proposed method. Experimental results show that the proposed method has higher detection accuracy compared to the traditional rumor detection method.

**Keywords:** false information detection; residual structure; graph neural network

## 1. Introduction

An important feature of the information age is the emergence of information, which includes a great deal of disinformation. This disinformation influences people's decision making and can trigger social conflict. With the spread of the internet, disinformation often comes in the form of online rumors. Online rumors usually refer to words spread through online communication media (such as Weibo, WeChat, forums, etc.), which have no basis in fact and have an offensive and purposeful nature [1]. Online rumors are often used for fraud and phishing, which pose a significant threat to the safety and interests of individuals and society, making the detection of false information increasingly important.

The application scenarios of false information detection are extensive, involving multiple fields such as news media, social networks, and e-commerce. In the news media field, false information detection can help news organizations and reporters distinguish and block fake news, improving the credibility of news reporting. In the social network field, false information detection can help social network platforms discover and remove false information in a timely manner, thereby maintaining the health and order of the network space. In the e-commerce field, false information detection can help consumers identify fake goods and false advertising and safeguard the rights and interests of consumers.

In false information detection tasks, there are many research results available for reference. Traditional content-based approaches, which analyze the credibility of individual tweets or claims separately, ignore the high correlation between tweets and events and do

not consider information from human–content interaction data. In recent years, many methods based on deep learning and methods based on novel feature fusion have also emerged. For example, Ma proposed a method for rumor detection using a tree-structured recursive neural network, and the results show that the proposed method can achieve excellent early detection of rumors [2]. However, this method may have some difficulties in processing long texts and complex syntax structures. Vu proposed a new model based on graph convolutional networks and propagation embedding for rumor detection in social media and conducted sufficient experiments on real datasets to prove the effectiveness of the method [3]. However, this method is aimed at text propagation in social networks and may not be suitable for cases involving visual or other non-textual information in the propagation process. In addition, Monti proposed a new type of automatic fake news detection model based on geometric deep learning, and experiments showed that social network propagation and structure are important features for the highly accurate detection of fake news [4]. However, this method may be affected by the sparseness and incompleteness of the data. Social media data often have high dynamicity and noise and lack complete information labels, which may lead to a decline in the performance of the model. In addition, since different news data are often interrelated, they are naturally interactive. Graph neural networks that are good at processing graph structures are also applied to rumor detection tasks. For example, Lotfi proposed a model that uses graph convolutional networks to detect rumor conversations, which extracts reply trees and user graphs for each conversation, achieving better performance compared to the baseline method [5]. Bian proposed a bidirectional graph model called a bidirectional graph convolutional network (Bi-GCN) to explore both features through top-down and bottom-up rumor propagation. It uses a GCN with a top-down directed graph of rumor propagation to learn the patterns of rumor propagation, and a GCN with a rumor propagation graph in the opposite direction to capture the structure of rumor propagation, which empirically demonstrated the superiority of this method over state-of-the-art methods [6]. Qian proposed a hierarchical multimodal contextual attention network (HMCAN) that uses the Resnet and Bert models to learn the features of images and text, respectively, and designed a hierarchical coding network to capture hierarchical semantics for fake news detection [7].

In contrast to existing methods, this article proposes a rumor detection method for the interaction characteristics of both the content and structure named GCRES, which can combine content and propagation patterns to characterize rumors. The residual structure, based on a graph convolutional network, is used to model the content structure correlations of heterogeneous graphs, thus overcoming the problem of the indistinguishable effects of adjacent neighbor nodes in the rumor propagation process and effectively realizing rumor classification. Specifically, the main contributions of this article include the following three points:

(1) This article constructs a heterogeneous graph to obtain a representation of rumors by combining post content and rumor propagation patterns, which includes the textual information of rumors and the initial propagation information.

(2) This article proposes a residual structure based on a graph convolutional network. This structure uses a skip connection method to effectively overcome the problem of the indistinguishable effects of adjacent nodes in the rumor propagation process and obtain the interaction characteristics of heterogeneous graphs.

(3) This article uses the Twitter15 and Twitter16 datasets, which are widely used in rumor detection for experimental verification. The experimental results show that, compared with traditional rumor monitoring methods, the proposed GCRES method can achieve a higher rumor detection accuracy.

## 2. Related Works

Traditional methods for detecting fake information can be categorized into two types: rule-based methods and machine learning-based methods. Rule-based methods classify real and fake information using the differences between them, including features like

keywords, sentence structures, and sentiment polarities in the text [8]. On the other hand, machine learning-based methods classify real and fake information by building models, such as using support vector machines, random forests, or other algorithms to train and classify data. Ma et al. first used deep learning models to detect rumors on Weibo [9]. In subsequent studies, they proposed two recursive neural network models based on top-down and bottom-up tree structures to better capture rumor structures and text features. The results showed that this model achieved high accuracy in detecting early propagating rumors [2].

Since introducing deep learning methods, the accuracy and efficiency of fake information detection have significantly improved. Among them, graph neural networks, as a powerful representation learning method, have a wide range of applications in fake information detection. Graph neural network-based fake information detection methods usually fall into two categories: node-based methods and graph-based methods.

Node-based methods mainly focus on node features and contextual information, whereas graph-based methods use the entire graph as input and utilize graph neural networks to learn graph representations. In node-based methods, the most commonly used approach is to use the social and content attributes of nodes for fake information detection, for example, by using features such as user information, text content [2,10], and time information to determine whether the node is spreading fake information. Y Liu et al. used recurrent and convolutional networks to construct a time-series classifier to capture global and local variations in user features on the propagation path, thus detecting fake news [11]. This method is the first to model the news dissemination path on social media as multi-dimensional time series and practice fake news detection through a sequence classifier. Ling Sun et al. discussed a novel joint learning model called HG-SL for the early detection of fake news. This model uses a hyper-GNN to embed the global relationships of users, and multi-head self-attention modules to simultaneously learn local contexts (local context in specific news) during propagation in order to comprehensively capture the differences between true and false news. The introduction of global node centrality and local propagation status further highlights user influence and news dissemination ability. The experiments show that HG-SL is significantly better than the SOTA models in the early detection of fake news [12]. In addition, some studies have also considered the propagation behavior of nodes as one of the node features, such as the forwarding and like counts of nodes [13,14].

In graph-based methods, the main approach is to learn the representation of the entire graph through graph neural networks and then perform fake information detection. For example, Tian Bian et al. proposed a novel bidirectional graph convolutional network (Bi-GCN) model, which uses a rumor propagation-directed graph with a top-down structure to learn the propagation patterns of rumors, and a rumor propagation graph with a reverse direction to capture the structure of rumor propagation. The influence of the original post of the rumor is enhanced in the graph structure, and the model achieved excellent results in fake information detection [6]. K Tu et al. proposed a framework for rumor representation learning and detection. This framework uses combined text and propagation structure representation learning to improve rumor detection performance. The authors proposed a joint graph concept to integrate the propagation structure of all tweets to alleviate the sparsity issue of the propagation structure in the early stage [15]. Some researchers have also combined graph neural networks with attention mechanisms to learn graph representations more accurately. For instance, Qi Huang et al. proposed a meta-path-based heterogeneous graph attention network framework. The heterogeneous graph is decomposed into tweet word and user subgraphs according to tweet words and tweet user paths, and node representations are learned using subgraph attention networks to capture the global semantic relationships of text content and fuse information involved in source tweet propagation for rumor detection [16]. Chunyuan Yuan et al. proposed a novel global-local attention network (GLAN) for rumor detection on social media. Their method combines local semantic relationships with global structural information, uses multi-head

attention mechanisms to integrate the semantic information of relevant retweets into the source tweet, generates a better-integrated representation, establishes a heterogeneous graph using global structural information to capture the complex global information between different source tweets, and uses global attention for rumor detection. Experimental results show that the GLAN is significantly better than existing models in rumor detection and early detection [17]. In addition, Ma et al. used statistical features in three aspects, including rumor content language characteristics, user characteristics participating in rumor transmission, and the propagation network structure to build a feature graph. The authors integrated entity recognition, sentence reconstruction, and ordinary differential equation networks into a unified framework called ESODE, which improved the performance of rumor detection [18].

## 3. Solution Design

### 3.1. System Model

A rumor detection dataset $X = \{x_1, x_2, \ldots x_n\}$, where $x_i$ represents the i-th event, and n represents the total number of events. Each event $x_i$ contains two independent sets: content and propagation structure. In addition, $x_i = \{P_i, G_i\}$, where $P_i$ and $G_i$ represent the content and propagation structure of $x_i$, respectively. The propagation structure $G_i$ is composed of a set of nodes $C_i = \{c_{i0}, c_{i1}, \ldots, c_{iN}\}$ and a set of connections $E_i = \{e_{st} \mid s, t = 0, 1, \ldots, N\}$, where $c_{i0}$ represents the original post, and $c_{ij}$ represents the $j$-th response post. Let $A \in \{0, 1\}^{N \times N}$ be the adjacency matrix, where $A_{st} = 1$ if there exists an edge from node $c_{is}$ to node $c_{it}$, and $A_{st} = 0$ otherwise. Moreover, Let $A'$ be the adjacency matrix of the heterogeneous graph that combines both the content and propagation structure. In this paper, we model the rumor detection task as a supervised classification problem, where each event has a true label $y_i$, with a value set of $\{NR, FR, TR, UR\}$, representing non-rumors, falsehood rumors, true rumors, and unverified rumors, respectively. Our goal is to train a classifier $f : x_i \to y_i$ to accurately predict the label of the content and the propagation structure of a given post.

### 3.2. Rumor Detection Framework

A heterogeneous graph is a graph in which there are multiple different types of nodes and multiple types of edges between the different types of nodes. Compared with traditional homogeneous graphs, heterogeneous graphs have richer structures and more associated information. In heterogeneous graphs, nodes are divided into different types, and there is a clear distinction between node types. For example, the heterogeneous composition of a social network may include user nodes, post nodes, and tag nodes. The edges between the different node types can represent the follow relationship between users, the interaction relationship between users and posts, and the association relationship between posts and tags. Such heterogeneous graphs can more accurately simulate the complex relationship network in the real world and capture rich correlation information between different nodes.

As shown in Figures 1 and 2, we propose a rumor detection framework based on the interaction characteristics of heterogeneous graphs, named GCRES. The framework consists of three parts: rumor representation, interaction representation learning, and rumor classification. Firstly, we construct a graph model based on the content and propagation structure for rumor representation. Specifically, we first use the TF-IDF model to extract the representation information of the rumor content [19]. Then, we encode the propagation node vectors using the adjacency matrix, combine the content and propagation information, and finally embed the joint graph into a low-dimensional space. Secondly, we use a graph convolutional network (GCN) to learn the initial state of the heterogeneous graph. At the same time, we use interaction representation learning technology to obtain the interaction features of the heterogeneous graph. Finally, we use an average pooling layer to cascade the features of heterogeneous maps and make predictions of rumor categories.

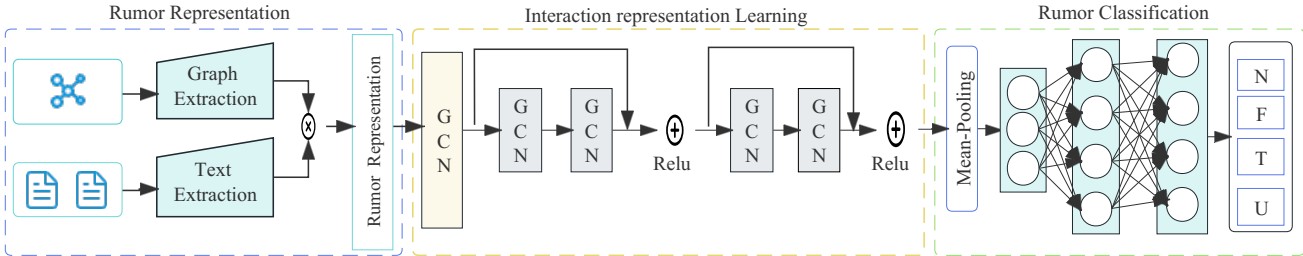

**Figure 1.** Rumordetection framework.

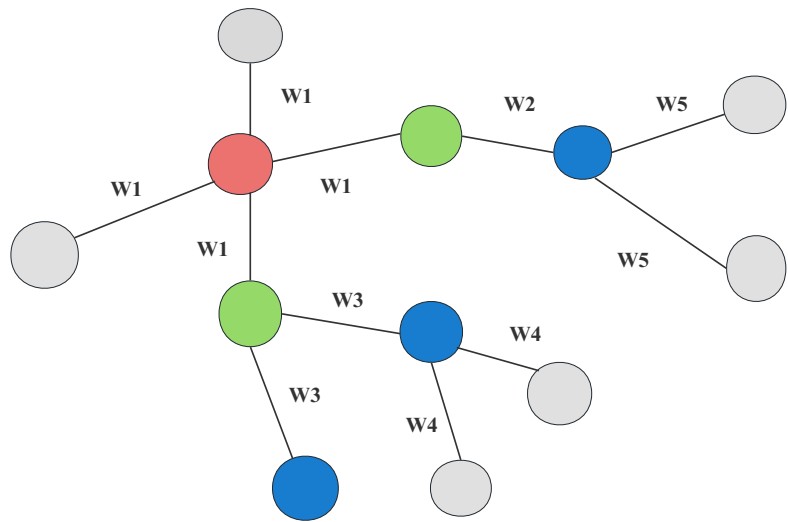

**Figure 2.** Heterogeneousgraph combining rumor content and propagation structure. The red circles represent the original nodes, and the other colored circles represent the nodes in the propagation path

### 3.3. Rumor Representation

As shown in Figure 1, we constructed a heterogeneous graph to characterize rumors by combining the content and propagation structure. Specifically, we used the TF-IDF method to obtain the representation of the content in each post. First, we filtered out stop words and constructed a corpus. Then, the word frequency can be represented as:

$$tf_{ij} = \frac{n_{ij}}{\sum_k n_{ik}} \tag{1}$$

In the equation, $n_{ij}$ represents the number of times the $i$-th vocabulary appears in the $j$-th post. At the same time, inverse document frequency can be represented as:

$$idf_i = \frac{\log |D|}{1 + |k : t_i \in p_k|} \tag{2}$$

where $|D|$ represents the total number of posts in the corpus, and $|k : t_i \in p_k|$ represents the number of posts containing the vocabulary $t_i$. Then, the weight of the vocabulary $t_i$ can be represented as:

$$t_i = tf_{ij} \times idf_i \tag{3}$$

Therefore, the content representation of the $j$-th post is $p_j = \left[t_1, t_2, \ldots, t_{|W|}\right]$, where $|W|$ is the total number of vocabulary in the corpus.

On the other hand, since the representation of $p_j$ is high dimensional and sparse, we use an embedding layer to map it to a low-dimensional space to obtain dense real-valued vectors as the content representation of the $j$-th post:

$$v_j = W_j p_j \tag{4}$$

where $W_j$ represents the weight of the embedding layer.

Based on the above framework, we further construct the propagation structure of the rumor $G_j = \{C_j, E_j, A\}$. Since the adjacency matrix $A \in \{0, 1\}^{N \times N}$ reflects the transmission path of the rumor, the adjacency matrix of the joint rumor content and propagation path can be obtained from $A$ and $v_j$:

$$A'_{st} = A_{st}v_j \tag{5}$$

where $A_{st}$ represents the weight of the edge from node $c_{js}$ to node $c_{jt}$. Based on this, the initial representation of the rumor $r_j$ can be represented as:

$$r_j = \frac{1}{M} \sum_{c_{js}, c_{jt} \in C_j} A'_{st} \tag{6}$$

where $M$ is the number of edges occupied by the $j$-th post in the propagation structure.

### 3.4. Acquiring the Initial State

After obtaining the representation of the rumor, we use a GCN to obtain the initial state of the heterogeneous graph. Specifically, we first construct the operator $\tilde{A}$ of the GCN, defined as:

$$\Phi = \widetilde{D}^{-\frac{1}{2}} \tilde{A} \widetilde{D}^{-\frac{1}{2}} \tag{7}$$

$$\tilde{A} = A' + I \tag{8}$$

where $A'$ and $I$ are the adjacency matrix and identity matrix of the heterogeneous graph, respectively, and $\widetilde{D}$ is the degree matrix of $\tilde{A}$. Then, the initial state representation of the joint graph can be represented as:

$$H^{(0)} = \sigma\left(\Phi r_j W^{(0)}\right) \tag{9}$$

where $H^{(0)}$ represents the initial state of the rumor propagation, $W^{(0)}$ is the weight matrix of the filter, and $\sigma(\cdot)$ is the ReLU activation function.

### 3.5. Interaction Representation Learning

In order to obtain the interaction feature information of the rumor content and propagation graph, we use interaction representation learning to learn the continuous temporal correlation in the propagation process.

First, we assume that the hidden representations of all nodes are correlated, and we combine the GCN and residual network (ResNet) methods to learn the embedding representations of the nodes. Specifically, we use a two-layer GCN network model to learn the representation of the propagation graph and extract potential relationships through the feature information of the nodes themselves, as well as the adjacent nodes. Each GCN layer first obtains the features of the current node and the adjacent nodes, then uses aggregation functions to obtain local feature relationships, and finally trains through shallow learning to obtain high-dimensional features. The propagation relationship between layers can be represented as:

$$H^{(l+1)} = \sigma\left(\widetilde{D}^{-\frac{1}{2}} \tilde{A} \widetilde{D}^{-\frac{1}{2}} H^{(l)} W^{(l)}\right) \tag{10}$$

where $\tilde{A} = A + I$, $I$ is the identity matrix, $\widetilde{D}$ is the degree matrix of $\tilde{A}$, and $\sigma(\cdot)$ is the ReLU activation function.

On the other hand, we use a structure based on residual networks to cascade four GCN networks. As the number of model layers in the GCN increases, it is prone to the problem of over-smoothing, which makes the differentiation of neighboring nodes' effects in the rumor propagation process unclear. Therefore, a graph neural network incorporating

residual networks is designed, adopting skip connection to improve and avoid this problem. As shown in Figure 1, the output of each residual block can be represented as:

$$y^{(l+1)} = h^{(l)}\left(x^{(l)}\right) + F\left(x^{(l)}, H^{(l)}\right) \tag{11}$$

*3.6. Rumor Classification*

We further use the average pooling operation to aggregate the output of all GCNs, that is,

$$H = \frac{1}{L} \sum_{l=0}^{L} H^{(l)} \tag{12}$$

where $L$ is the number of GCNs. Then, we use fully connected layers and a softmax layer for rumor classification, which is:

$$\hat{y} = \text{softmax}(W_{FC} H + b_{FC}) \tag{13}$$

where $W_{FC}$ and $b_{FC}$ are the weight and bias of the last hidden layer, and $\hat{y}$ represents the predicted probability vector of rumors belonging to each category, which is also the label for predicting rumor events. Finally, we train the model by minimizing the cross-entropy between $\hat{y}$ and the true distribution $y$ and adding L2 regularization to prevent overfitting.

## 4. Experiment

*4.1. Dataset*

We conducted experimental validation of the proposed method using two open source datasets, Twitter15 and Twitter16. Both datasets consist of social media data, each with 1490 and 818 propagation graphs. The labeling of the propagation graphs includes four types: non-rumor, true rumor, false rumor, and unverified rumor. In addition, the nodes in the graph represent users, and the edges represent replies and forwarding. Table 1 provides statistical information on the two datasets.

**Table 1.** Twitter15 and Twitter16 datasets.

| Statistical Information | Twitter15 | Twitter16 |
|---|---|---|
| Total number of posts | 331,612 | 204,820 |
| Original post count | 1490 | 818 |
| Non-rumor count | 372 | 205 |
| True rumor count | 374 | 203 |
| False rumor count | 370 | 205 |
| Unverified rumor count | 374 | 205 |

*4.2. Baseline Methods*

We used the following seven baseline methods for rumor detection:

(1)  DTC [10]: This method uses decision tree classifiers and manually designed features to extract and analyze tweet information.
(2)  RFC [20]: This method uses a random forest classifier to detect rumors by combining user features, language features, and news structure features.
(3)  SVM-TS [12]: This method is based on an SVM classifier, using manually designed features to form a time-series kernel to identify rumors.
(4)  SVM-HK [21] : This method uses a graph kernel to measure the similarity of propagation structures combined with an SVM classifier for rumor detection.
(5)  GRU-RNN [9]: This method relies on a recurrent neural network with GRUs to capture the contextual changes in relevant posts over time for rumor detection.
(6)  BU-RvNNand TD-RvNN [22]: This method adopts a bidirectional tree-structured recursive neural model, which includes top-down and bottom-up tree-structured neural networks combined with GRUs to learn and analyze rumor information.

(7)  Rumor2vec [15]: This method uses a CNN-based model to combine the textual content with the propagation structure to achieve joint representation learning for rumor detection.

### 4.3. Experimental Setup

We used a TF-IDF model based on a 5000-word vocabulary to represent the content of the post. The node and hidden layer embedding sizes were searched in $\{32, 64, 128, 256\}$. The embedding size of the model and the number of samples selected for each training were both set to 64. At the same time, we divided each dataset into five parts and performed fivefold cross-validation to ensure the robustness and fairness of the experimental results. We used the accuracy values for the four categories and the F1 value for each category as performance indicators. In addition, we used the Adam algorithm for model optimization. The learning rate was 0.005, and the number of iterations was 100 epochs. Finally, to prevent overfitting, we stopped training early when the validation stopped decreasing for 10 epochs.

### 4.4. Analysis of the Results

We used the accuracy and F1 values to evaluate the performance of our proposed classification model. The accuracy rate helps us understand the accuracy of the classification, whereas the F1 value takes into account the accuracy and recall, allowing us to more comprehensively evaluate the classification effect of the model on different classes. In Table 2, Acc. represents accuracy, and NR F1, TR F1, FR F1, and UR F1 represent the F1 values of the non-rumor, true rumor, false rumor, and unverified rumor categories.

**Table 2.** Comparison experiment results.

| Method | Acc. | NR F1 | FR F1 | TR F1 | UR F1 |
|---|---|---|---|---|---|
| Twitter15 dataset | | | | | |
| DTC | 0.454 | 0.733 | 0.355 | 0.317 | 0.415 |
| RFC | 0.565 | 0.810 | 0.422 | 0.401 | 0.543 |
| SVM-TS | 0.544 | 0.796 | 0.472 | 0.404 | 0.483 |
| SVM-HK | 0.493 | 0.650 | 0.439 | 0.342 | 0.336 |
| GRU-RNN | 0.641 | 0.684 | 0.634 | 0.688 | 0.571 |
| BU-RvNN | 0.708 | 0.695 | 0.728 | 0.759 | 0.653 |
| TD-RvNN | 0.723 | 0.682 | 0.758 | 0.821 | 0.654 |
| Rumor2vec | 0.796 | 0.883 | 0.746 | 0.836 | 0.723 |
| GCRES | 0.853 | 0.855 | 0.858 | 0.903 | 0.746 |
| Twitter16 dataset | | | | | |
| DTC | 0.473 | 0.254 | 0.080 | 0.190 | 0.482 |
| RFC | 0.585 | 0.752 | 0.415 | 0.547 | 0.563 |
| SVM-TS | 0.574 | 0.755 | 0.420 | 0.571 | 0.526 |
| SVM-HK | 0.511 | 0.648 | 0.434 | 0.473 | 0.451 |
| GRU-RNN | 0.633 | 0.617 | 0.715 | 0.577 | 0.527 |
| BU-RvNN | 0.718 | 0.723 | 0.712 | 0.779 | 0.659 |
| TD-RvNN | 0.737 | 0.662 | 0.743 | 0.835 | 0.708 |
| Rumor2vec | 0.852 | 0.857 | 0.769 | 0.927 | 0.850 |
| GCRES | 0.888 | 0.801 | 0.877 | 0.912 | 0.919 |

Table 2 shows the test results of the proposed method and the seven baseline methods on Twitter15 and Twitter16. In the table, it can be seen that compared with methods using manually designed features (DTC, RFC, SVM-TS, and SVM-HK), methods based on deep learning (GRU-RNN, BU-RvNN, Rumor2vec) and the proposed method achieved higher accuracy and F1 values. This is because deep learning technology can learn effective high-dimensional features of rumors. These results prove that deep learning technology can effectively improve the performance of rumor detection.

On the other hand, the proposed method achieved the highest accuracy on both datasets. Specifically, the proposed method achieved accuracies of 85.3% and 88.8% on Twitter15 and Twitter16, respectively, which were 5.7% and 3.6% higher than the baseline method Rumor2vec. This is because CNNs cannot handle data with dynamic features, making it difficult for Rumor2vec to capture the interaction characteristics of rumor propagation. These results demonstrate that incorporating interaction features can effectively improve the performance of rumor detection. It is worth noting that for the non-rumor category, Rumor2vec outperformed the proposed method. This is because non-rumor posts are often posted or replied to by more authoritative and credible users, so the propagation path and content of non-rumor posts are more fixed. Therefore, the interaction characteristics of non-rumor posts have little impact on the detection performance.

Figures 3 and 4 depict the confusion matrices of the experimental results for the Twitter15 and Twitter16 datasets, respectively. As shown in Figure 3, for the Twitter15 dataset, the accuracy of our method was high for data that were identified as true rumors or false rumors. But the accuracy achieved for unverified rumors was only 0.746 due to the peculiarities of the data in this category. As they were unverified, the samples in this category could be both true or false, which is very confusing. Therefore, our method is more suitable for only facticity and falsehood identification of known data, and for unidentified rumors, there are no features that can be relied on from the data.

As shown in Figure 4, for the Twitter16 dataset, true rumors, false rumors, and unverified rumors were more accurately identified compared to non-rumors. The reason for this may lie in the correlation between the rumor sample data and whether a sample was a true rumor or a false rumor, as some characteristics can be discerned from a textual point of view (unverified rumors may be true or false). However, for the non-rumor sample, it was less related to the other three categories, so its classification accuracy was low.

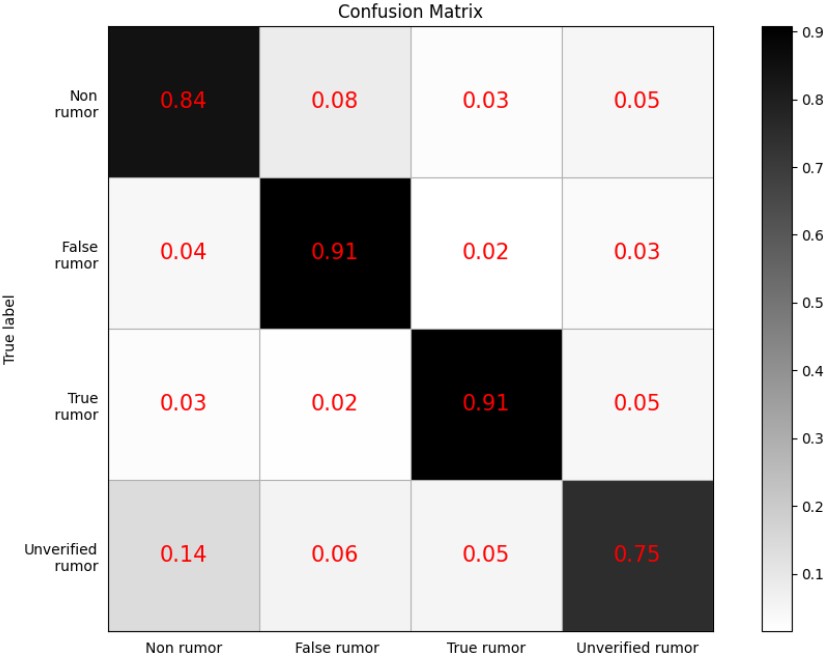

**Figure 3.** Confusion matrix of Twitter15.

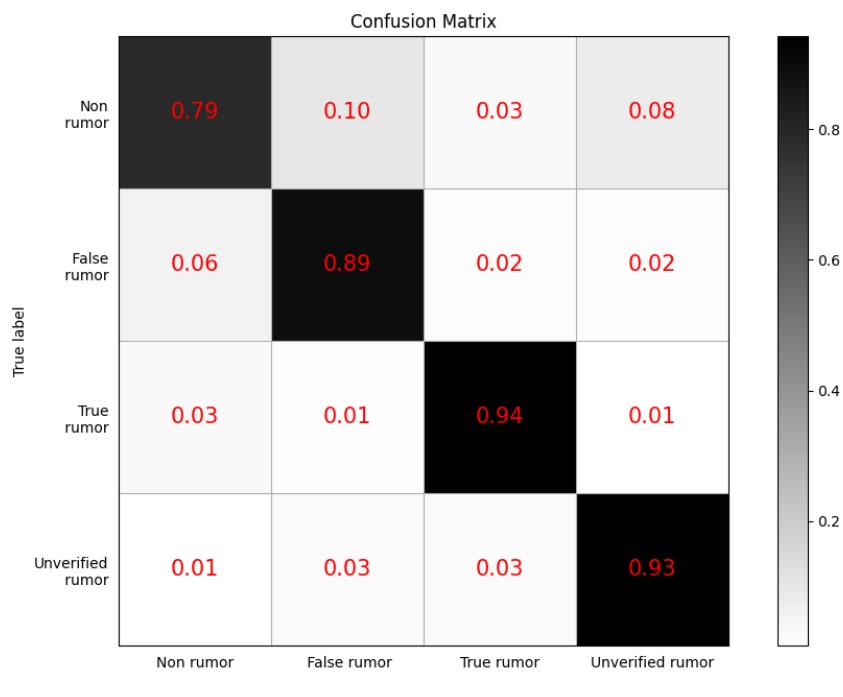

**Figure 4.** Confusion matrix of Twitter16.

## 5. Conclusions

By utilizing the dynamic changes in rumor content, propagation structure, and propagation heterogeneous graph, this paper proposes a novel interaction rumor detection method based on a graph convolutional network called GCRES. Firstly, a novel heterogeneous graph is proposed by combining the rumor content and propagation structure to obtain accurate rumor representation. Secondly, a residual module consisting of four cascaded graph convolutional networks is designed using the powerful ability of graph convolutional networks in dealing with heterogeneous graphs for representation learning, thus fully mining the interaction characteristics of the heterogeneous graph. Experimental results show that compared with traditional rumor detection methods, the GCRES method exhibits better detection performance on both the Twitter15 and Twitter16 datasets.

**Author Contributions:** Conceptualization, D.Y. and Y.Z.; methodology, Y.Z. and N.Y.; software, N.Y. and S.Z.; validation, S.Z. and Y.Z.; formal analysis, N.Y. and S.Z.; writing—original draft preparation, N.Y., Y.Z., and D.Y.; writing—review and editing, Y.Z. and K.-k.S.; visualization, N.Y. and K.-k.S.; supervision, K.-k.S.; funding acquisition, D.Y. All authors have read and agreed to the published version of the manuscript.

**Funding:** This work is supported by the National Social Science Funds of China (Grant No. 22BSH025), National Natural Science Foundation of China (Grant No. 61803047), Major Project of The National Social Science Foundation of China (19ZDA149, 19ZDA324) and Fundamental Research Funds for the Central Universities (14370119, 14390110). Ke-ke Shang is supported by Jiangsu Qing Lan Project.

**Institutional Review Board Statement:** Not applicable.

**Informed Consent Statement:** Not applicable.

**Data Availability Statement:** Not applicable.

**Acknowledgments:** College of Media Engineering, Communication University of Zhejiang; Institute of Intelligent Media Technology, Communication University of Zhejiang; Key Lab of Film and TV Media Technology of Zhejiang Province; Computational Communication Collaboratory, Nanjing University.

**Conflicts of Interest:** The authors declare no conflict of interest

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
