# Peer review of "Graph Convolutional-Based Deep Residual Modeling for Rumor Detection on Social Media"

_mathematics, doi:10.3390/math11153393_

Round 1
Reviewer 1 Report
This is an interesting and well-written paper on application of deep learning in rumor classification.
I have the following minor concerns:
1. the Introduction section is too shalow and covers only three reserches from the last five years. Please prepare a more elaborate state of the art.
2. Is a rumor detection defined as a binary classification problem?
3. please define a rumor in a formal way
4. section 3 - "A Rumor Detection Dataset X" and subsequent sentences - are these definitions?
If so, please format the sentences appropriately and use the full equations, as they are currently unintelligible.
Please add the numbers to the equations.
f:xi -> yi - yi is not defined anywhere.
5. 3.5 - "Firstly, First," please correct.
6. "On the other hand, this article uses a structure based on residual networks to cascade
four GCN networks. " - which article? Do you mean a method presented in this paper or an online article with potential rumor?
7. table 2 - is Acc. accuracy? Please describe all abbreviations used in the table.
8. please conduct a confusion matrix analysis of the results.
Please identify which data examples the algorithm has the most difficulty with in order to make a correct classification.
Can this be improved somehow?
9. please publish implementation details and source codes to make experiments reproducible.
Author Response
Dear Editor, Thank you for allowing a resubmission of our manuscript, with an opportunity to revise manuscript comments provided by reviewers. We are uploading (a) our point-by-point response to the comments (below) (response to reviewers), and (b) a clean updated manuscript.
Answer to reviewers:
Review 1:
This is an interesting and well-written paper on application of deep learning in rumor classification.
I have the following minor concerns:
- the Introduction section is too shallow and covers only three researches from the last five years. Please prepare a more elaborate state of the art.
Author Response:
Dear reviewer, thank you very much for your comments and valuable suggestions. In the revised introduction, we add some of the latest works in the art on graph neural networks-based and multimodal-based rumor detection. With these detailed status quo descriptions, we will be able to provide a more comprehensive overview of the latest developments in the field of rumor detection and ensure that the introduction section has sufficient depth and breadth. The additional content is as follows:
“Traditional content-based approaches, which analyze the credibility of individual tweets or claims separately, ignore the high correlation between tweets and events, and do not take advantage of information from human-content interaction data. In recent years, many methods based on deep learning and methods based on novel feature fusion have also emerged.”
“In addition, since different news data are often interrelated, they are naturally interactive. Graph neural networks that are good at processing graph structures are also applied to rumor detection tasks, such as Lotfi proposed a model that uses graph convolutional networks to detect rumor conversations, extracting reply trees and user graphs for each conversation, which has better performance for the baseline method[4]. Bian proposes a bidirectional graph model called bidirectional graph convolutional network (Bi-GCN) to explore both features through top-down and bottom-up rumor propagation. It uses GCN with a top-down directed graph of rumor propagation to learn patterns of rumor propagation; and GCN with a rumor propagation graph in the opposite direction to capture the structure of rumor propagation, which empirically demonstrates the superiority of this method over state-of-the-art methods[5]. Qian proposes a hierarchical multimodal contextual attention network (HMCAN) that uses the Resnet model and the Bert model to learn the features of images and text, respectively, and designs a hierarchical coding network to capture hierarchical semantics for fake news detection[6].”
- Is a rumor detection defined as a binary classification problem?
Author Response:
Rumor detection can often be defined as a binary classification problem whose goal is to divide a given text into two categories: rumors and non-rumors. For each piece of text, the algorithm needs to determine whether it is a rumor or a non-rumor. But in some datasets, rumor detection problems can also be defined as multiclassification problems, such as dividing text into categories such as "rumor", "truth", and "uncertainty" in order to describe the nature of the rumor in more detail. The datasets Twitter 15 and Twitter 16 used in this study included four categories: Non-rumor, True rumor, False rumor, and Unverified rumor, some of which were samples that were unproven rumors, and in fact they could be rumors or non-rumors. These samples present certain challenges to the rumor detection task.
- please define a rumor in a formal way
Author Response:
According to Allport and Bosman, the pioneers of rumor research, rumors are: "propositions related to the events of the time that are believed are generally circulated among people in oral media, but lack specific information to confirm their accuracy" According to researcher Knapp, rumors are "declarations designed to make people believe, related to current events, and widely circulated without official confirmation." Considering that the research in this paper mainly focuses on social media network rumors, therefore, in this paper, we quote researcher Jia J W's definition of online rumors: "Internet rumors usually refer to words spread through online communication media (such as Weibo, WeChat and forums, etc.), which have no factual basis and have a certain offensive and purposeful nature." We have added the above definition of online rumors to the introduction of the article. The additional changes are as follows:
“An important feature of the information age is the emergence of information, which includes a great deal of disinformation. This disinformation influences people's decision-making and triggers social conflict. With the spread of the internet, disinformation often comes in the form of online rumors. Online rumors usually refer to words spread through online communication media (such as Weibo, WeChat and forums, etc.), which have no basis in fact, and have a certain offensive and purposeful nature[1]. Online rumors are often used for fraud and phishing, which pose a significant threat to the safety and interests of individuals and society, making the detection of false information increasingly important.”
- section 3 - "A Rumor Detection Dataset X" and subsequent sentences - are these definitions?
If so, please format the sentences appropriately and use the full equations, as they are currently unintelligible. Please add the numbers to the equations. f:xi -> yi - yi is not defined anywhere.
Author Response:
"A rumor detection dataset X" and its subsequent sentences are not definitions, and this paragraph is intended to help readers understand the systematic model of this article. Among them, the definition of yi is given in subsection 3.1 of the text: "In this paper, we model the rumor detection task as a supervised classification problem, where each event has a true label yi"( yi represents the true classification label of each event)
- 5 - "Firstly, First," please correct.
Author Response:
Thanks for correcting. In the revised version, we have removed the "First" repetition word, carefully checked the language of the article, and made corresponding changes to the expressions that do not conform to the grammar.
- "On the other hand, this article uses a structure based on residual networks to cascade
four GCN networks. " - which article? Do you mean a method presented in this paper or an online article with potential rumor?
Author Response:
We apologize for the lack of clarity in this article. The paper states that "on the other hand, this paper uses a residual network-based structure to cascade four GCN networks. Referring to the Interaction representation Learning section proposed in this paper, we design a GCN architecture for fusion residual networks, as shown in the middle section of Figure 1. Hop connections are used to improve and avoid the problem of excessive smoothing that tends to occur when the number of GCN layers increases.
In order to avoid ambiguity caused by unclear expressions, in the revised draft, we have put "this article uses a structure based on residual networks to cascade four GCN networks." Modified to "We use a structure based on residual networks to cascade four GCN networks."
- table 2 - is Acc. accuracy? Please describe all abbreviations used in the table.
Author Response:
Thanks for the suggestion. We have read and noticed the question you mentioned about the use of abbreviations. Yes, Acc stands for Accuracy. Based on your suggestions, we have described these abbreviations accurately in Section 4.4 of the revised draft. The specific contents are as follows:
"We used the accuracy and F1 values to evaluate the performance of our proposed classification model. The accuracy rate helps us understand the accuracy of the classification, while the F1 value takes into account the accuracy and recall more comprehensively, helping us evaluate the classification effect of the model on different classes. In Table 2, Acc represents accuracy, NR F1, TR F1, FR F1, UR F1 represent the F1 values of the categories Non rumor, True rumor, False rumor, and Unverified rumor.”
- please conduct a confusion matrix analysis of the results.Please identify which data examples the algorithm has the most difficulty with in order to make a correct classification.Can this be improved somehow?
Author Response:
A confusion matrix analysis of the results has been added to section 4.4 of the revised version with the following additions:
“Figures 3 and 4 are the experimental results confusion matrix for the datasets Twitter15 and Twitter16, respectively. As shown in Figure 3, in the Twitter15 dataset, the accuracy of our method is high for data that has been identified as true rumors or false rumors. But the accuracy of only 0.746 for the Unverified rumor, is due to the peculiarities of the data in this category: it is unverified, and the samples under this category can actually be both true and false. It is very confusing. Therefore, our method is more suitable for only facticity and falsehood identification of unknown data, and for unidentified rumors, there are no features that can be relied on from the data.
As shown in Figure 4, in the Twitter16 dataset, true rumors, false rumors, and Unverified rumors are more accurate to identify and less accurate for Non rumor. The reason for this may lie in the correlation between the rumor sample data, and whether a sample is a true rumor or a false rumor, there are characteristics that can be used from a textual point of view(undefined rumors may be true or false). However, for the Non rumor sample, it is less related to the other three categories, so its classification accuracy is low.”
- please publish implementation details and source codes to make experiments reproducible.
Author Response:
Implementation details are described in Section 4.3 of the revised draft, and the source code has been sent to the editorial office along with the revised draft and the review comments.

Reviewer 2 Report
The authors proposed a method called GCRES which constructs a heterogeneous map of joint rumor content and propagation structure combining content and propagation patterns to obtain rumor representation. And then leveraging residual structure base on graph convolution network, the model learns interaction of structure and content. The paper is overall well written and the authors conducted thorough experiments and the analysis is well performed. However, several points need to be considered to clarify the significance of this paper as listed below:
1. The author proposed a heterogeneous map, but the information of heterogeneous graph is not very clear. A further explanation of what is heterogeneous map is recommended to make the idea clear.
2. The introduction is limited in terms of explanation. The first paragraphs of the introduction try to introduce the background, but in my opinion, it is quite shallow. The author needs to thoroughly explain the problem by adding some examples too if possible.
3. There are many technical issues in writing. Reorganization of the complete paper is needed. The language used in the sentences should be more clearer and to the point.
4. Author needs to be careful while using punctuations within sentences and checking the entire paper.
The authors proposed a method called GCRES which constructs a heterogeneous map of joint rumor content and propagation structure combining content and propagation patterns to obtain rumor representation. And then leveraging residual structure base on graph convolution network, the model learns interaction of structure and content. The paper is overall well written and the authors conducted thorough experiments and the analysis is well performed. However, several points need to be considered to clarify the significance of this paper as listed below:
1. The author proposed a heterogeneous map, but the information of heterogeneous graph is not very clear. A further explanation of what is heterogeneous map is recommended to make the idea clear.
2. The introduction is limited in terms of explanation. The first paragraphs of the introduction try to introduce the background, but in my opinion, it is quite shallow. The author needs to thoroughly explain the problem by adding some examples too if possible.
3. There are many technical issues in writing. Reorganization of the complete paper is needed. The language used in the sentences should be more clearer and to the point.
4. Author needs to be careful while using punctuations within sentences and checking the entire paper.
Author Response
Dear Editor, Thank you for allowing a resubmission of our manuscript, with an opportunity to revise manuscript comments provided by reviewers. We are uploading (a) our point-by-point response to the comments (below) (response to reviewers), and (b) a clean updated manuscript.
Answer to reviewers:
Review 2
The authors proposed a method called GCRES which constructs a heterogeneous map of joint rumor content and propagation structure combining content and propagation patterns to obtain rumor representation. And then leveraging residual structure base on graph convolution network, the model learns interaction of structure and content. The paper is overall well written and the authors conducted thorough experiments and the analysis is well performed. However, several points need to be considered to clarify the significance of this paper as listed below:
- The author proposed a heterogeneous map, but the information of heterogeneous graph is not very clear. A further explanation of what is heterogeneous map is recommended to make the idea clear.
Author Response:
Dear reviewer, thank you very much for your comments and valuable suggestions. We have noted the lack of explanation of heterogeneous graphs in the text you pointed out, and have added this part to section 3.2 of the article, which explains heterogeneous graph as follows:
"Heterogeneous graphs is a graph in which there are multiple different types of nodes, and there are multiple types of edges between different types of nodes. Compared with traditional homogeneous graphs, heterogeneous graphs have richer structure and more associated information. In heterogeneous graphs, nodes are divided into different types, and there is a clear distinction between node types. For example, the heterogeneous composition of a social network may include user nodes, post nodes, and tag nodes, and the edges between different node types can represent the following relationship between users, the interaction relationship between users and posts, and the association relationship between posts and tags. Such heterogeneous graphs can more accurately simulate the complex relationship network in the real world and capture rich correlation information between different nodes.”
- The introduction is limited in terms of explanation. The first paragraphs of the introduction try to introduce the background, but in my opinion, it is quite shallow. The author needs to thoroughly explain the problem by adding some examples too if possible.
Author Response:
Dear reviewer, thank you very much for your comments and valuable suggestions. In the revised introduction, we add some of the latest works in the art on graph neural networks-based and multimodal-based rumor detection. With these detailed status quo descriptions, we will be able to provide a more comprehensive overview of the latest developments in the field of rumor detection and ensure that the introduction section has sufficient depth and breadth. In addition, with the comments of another reviewer, we have also added the definition of online rumors to the introduction of the article. The additional changes are as follows:
“An important feature of the information age is the emergence of information, which includes a great deal of disinformation. This disinformation influences people's decision-making and triggers social conflict. With the spread of the internet, disinformation often comes in the form of online rumors. Online rumors usually refer to words spread through online communication media (such as Weibo, WeChat and forums, etc.), which have no basis in fact, and have a certain offensive and purposeful nature[1]. Online rumors are often used for fraud and phishing, which pose a significant threat to the safety and interests of individuals and society, making the detection of false information increasingly important.”
“Traditional content-based approaches, which analyze the credibility of individual tweets or claims separately, ignore the high correlation between tweets and events, and do not take advantage of information from human-content interaction data. In recent years, many methods based on deep learning and methods based on novel feature fusion have also emerged.”
“In addition, since different news data are often interrelated, they are naturally interactive. Graph neural networks that are good at processing graph structures are also applied to rumor detection tasks, such as Lotfi proposed a model that uses graph convolutional networks to detect rumor conversations, extracting reply trees and user graphs for each conversation, which has better performance for the baseline method[4]. Bian proposes a bidirectional graph model called bidirectional graph convolutional network (Bi-GCN) to explore both features through top-down and bottom-up rumor propagation. It uses GCN with a top-down directed graph of rumor propagation to learn patterns of rumor propagation; and GCN with a rumor propagation graph in the opposite direction to capture the structure of rumor propagation, which empirically demonstrates the superiority of this method over state-of-the-art methods[5]. Qian proposes a hierarchical multimodal contextual attention network (HMCAN) that uses the Resnet model and the Bert model to learn the features of images and text, respectively, and designs a hierarchical coding network to capture hierarchical semantics for fake news detection[6].”
- There are many technical issues in writing. Reorganization of the complete paper is needed. The language used in the sentences should be more clearer and to the point.
Author Response:
Thank you for your valuable comments. In the revised draft, we carefully reviewed the technical issues you pointed out in the writing, revised multiple sentences throughout the paper to ensure clarity and conciseness, and ultimately improved the overall quality, standardization, and readability of the paper. In addition, we have combined the opinions of the two reviewers to adjust and supplement the structure and content of the article to make the article smoother and clearer.
4) Author needs to be careful while using punctuations within sentences and checking the entire paper.
Author Response:
We have carefully reviewed the use of punctuation within sentences in the text.and revised many punctuation points that do not meet the specifications in the revised draft, thank you for your efforts to review our manuscript, and we hope that the revised paper meets your expectations.

Round 2
Reviewer 1 Report
Authors have addressed all my remarks. In my opinion paper can be accepted as it is.